# Consensus Learning with Deep Sets for Essential Matrix Estimation

**Dror Moran**          **Yuval Margalit**          **Guy Trostianetsky**

**Fadi Khatib**          **Meirav Galun**          **Ronen Basri**

Department of Computer Science and Applied Mathematics
Weizmann Institute of Science

## Abstract

Robust estimation of the essential matrix, which encodes the relative position and orientation of two cameras, is a fundamental step in structure from motion pipelines. Recent deep-based methods achieved accurate estimation by using complex network architectures that involve graphs, attention layers, and hard pruning steps. Here, we propose a simpler network architecture based on Deep Sets. Given a collection of point matches extracted from two images, our method identifies outlier point matches and models the displacement noise in inlier matches. A weighted DLT module uses these predictions to regress the essential matrix. Our network achieves accurate recovery that is superior to existing networks with significantly more complex architectures.

## 1 Introduction

Estimating the relative pose of two cameras depicting a stationary scene is a fundamental computer vision task and a basic step in multiview structure from motion (SFM) [33, 24, 45, 1, 51, 43, 20, 26, 35] and simultaneous localization and mapping (SLAM) [22, 47, 7, 21] pipelines. Both classical and recent deep network-based algorithms (see a review in Section 2) use point matches to compute the essential matrix, which encodes the relative position and orientation of the two cameras. Identifying such point matches by existing heuristics, however, is prone to mistakes, due to possibly large viewpoint changes, illumination differences, and the presence of ambiguous repetitive scene structures, resulting in *noisy matches* and extremely large numbers of *outlier matches* (often as many as 95%) that must be identified and pruned to enable accurate pose recovery.

Classical SFM algorithms use RANSAC [16] to robustly identify inliers and estimate pose parameters. While RANSAC has been used effectively for consensus recovery, learning-based deep network approaches have introduced a competitive alternative, making steady progress in accuracy while allowing for efficient inference and demonstrating resilience to very large fractions of outliers. This progress was obtained at the price of complicating the network architecture, e.g., using message passing in local, near-neighbour graphs [58, 28, 31, 52] or expensive attention (transformer) layers [28, 52], along with the addition of hard pruning steps [58, 28, 52].

In this paper, we introduce a simpler network architecture for consensus learning based on the Deep Sets framework [55]. Deep Sets architectures are based on shared, element-wise layers that are combined with global features produced by summing the element-wise features. Zaheer et al. and others [55, 50] proved that such architectures can express universal permutation-equivariant functions over sets. In our network, the input set elements include pairs of *keypoints*, i.e., the coordinates of matching pairs of points. In each layer, element-wise features are produced by a linear layer

38th Conference on Neural Information Processing Systems (NeurIPS 2024).

with shared weights, followed by SoftPlus activation. Global features are obtained by averaging the element-wise features, where averaging is used to maintain invariance to set cardinality. The network utilizes a stack of such permutation equivariant layers to classify point matches as either inliers or outliers and identify a consensus set to enable accurate relative camera pose regression. We further improve accuracy by integrating a noise regression module that aims to predict the displacement, due to noise, of the (clean) positions of inlier keypoints. Finally, we observe that training in two stages, i.e., first on a noise-free version of the real data (while including the outliers) and subsequently on the original real data, improves the accuracy of the predicted pose. Our network achieves accurate pose recovery that is superior to existing networks with significantly more complex architectures.

In summary, our contributions include:

- NACNet, a Noise Aware Consensus Network, for consensus learning tasks and robust geometric model estimation.
- A DeepSets based architecture that includes inlier displacement error estimation.
- An effective noise-free pretraining scheme: first, pretrain on a denoised version of the real data, then train on the real (noisy) data.
- Experiments demonstrate that NACNet achieves superior results compared to baselines on indoor and outdoor image pairs applied on various descriptors.

Our code is available at `https://github.com/drormoran/NACNet`.

## 2   Related work

**Classical methods.** RANSAC [16] and its successors, LO-RANSAC [8], USAC [39], MAGSAC [3], and MAGSAC++ [4] search over minimal point configurations to find consensus sets from noisy and corrupted data and estimate a corresponding parametric model. These are applied to matched keypoints with distinct descriptors obtained by filtering with Lowe's ratio test [30]. These classical methods are regarded as the standard solutions for finding consensus in data consisting of mixtures of inlier and outlier point matches.

**Learning-based methods.** Deep learning-based methods have been used recently to regress a geometric model and outlier classification. DFE [40] used a deep-based iteratively reweighted least squares (IRLS) scheme to predict inlier/outlier scores. LFGC [34] utilized an architecture that involves an inlier/outlier classifier and weight sharing, followed by context normalization, and applied a geometric loss to the output of the weighted 8-point algorithm (also called weighted DLT [17]).

Follow-up works improve prediction results by introducing more complex network designs. OANet [56] introduced an order-aware block, which contains differentiable permutation invariant pooling and unpooling operators that capture local context by utilizing soft clustering of correspondences in the feature space. CLNet [58] used this order-aware block together with pruning and local-to-global consensus learning procedure strategy to classify the correspondences by employing convolutions on local and global graphs built based on the Euclidean distance in feature space. All the methods mentioned above suffer from the leakage of outliers to the consensus set. Consequently, they all use RANSAC at the end of their inference step. In contrast, NCMNet [28], MGNet [31], and BCLNet [52] used weighted DLT also at inference, showing that the performance of the results is not improved further when RANSAC is applied in addition. NCMNet [28] proposed a local-to-global consensus learning scheme in which it first creates a local spatial graph, then a local feature space graph, and finally a global graph based on the inlier scores from the local graphs. BCLNet [52] introduced the idea of Bilateral Consensus, adopting the local graph from CLNet [58] as their projection step in a channel-wise transformer that learns global consensus. MGNet [31] used a similar scheme, building both implicit and explicit local graphs and a global graph. Unlike previous methods, this method does not prune correspondences inside the network.

In contrast to these methods, we use an architecture based on Deep Sets [55]. Deep Sets enable efficient information transfer between the point matches through global features without the need to construct and manipulate graphs. Our newly proposed noise regression module further improves our results. Finally, as with recent methods, our method too does not require a final RANSAC step.

**Learned feature matching.** Deep learning-based detectors and descriptors [11, 54, 13, 14] based on both CNNs and Transformers have been used in recent years to replace the handcrafted features

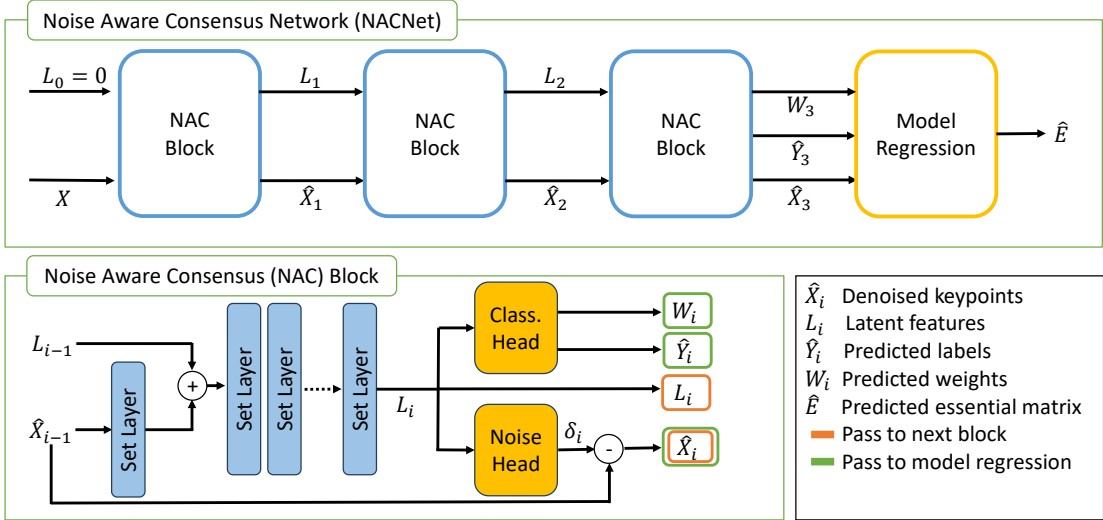

Figure 1: **Network architecture**. Noise Aware Consensus Network (NACNet) architecture, see text for details.

[30, 36, 5] used in classical methods. Those methods trained with challenging and diverse data have improved the accuracy and robustness of matching even with the classical nearest neighbor matching. Yet the main problems of high outlier rate remained. Consequently, learned matchers that match keypoints while rejecting non-matchable ones have merged[42, 27], combining Transformers with optimal transport[37] to produce more accurate matches even with large camera movement. These matchers rely on the descriptors for the matching and keypoint rejection and require RANSAC as a post-processing step. In contrast, our method, similar to [28], [31], [52], gets as an input the keypoints (point correspondences) only and does not incorporate RANSAC.

**Keypoint refinement.** Previous works [15, 48, 12, 26] have shown that correcting keypoints position could positively influence the results of geometric model estimation. All of those works use visual and learned features (SIFT descriptors or features obtained from applying a convolutional network to the input images) to correct the positions. To our knowledge, our paper is the first to use the input point set directly to correct keypoint position, as opposed to previous works, which rely on either an estimated geometric model or visual features.

## 3   Method

Consider a pair of images captured by (internally) calibrated cameras expressed with $3 \times 4$ matrices, $P = [I, \mathbf{0}]$ and $\tilde{P} = [R, \mathbf{t}]$, where $R$ and $\mathbf{t}$ respectively denote the relative rotation and translation between the two views. The essential matrix $E = [\mathbf{t}]_\times R$, determines the epipolar geometry between the two views, so that for any two corresponding points, $\mathbf{p}$ and $\tilde{\mathbf{p}}$, projected from a 3D point, it holds that $\tilde{\mathbf{p}}^T E \mathbf{p} = 0$. Existing algorithms commonly estimate the essential matrix directly from a set of putative matches between the two views, i.e., pairs of keypoints.

Our aim in this work is to construct a network that identifies a consensus set of point matches (a set of inliers), given a set of putative matches as input (generally contaminated with noise and outliers), and, based on this consensus set, predicts the essential matrix between the two images. We seek to construct a network that can overcome positional noise, which can reside in the inlier matches, and cope with a considerable fraction of outlier matches, up and above 95%. In addition, we aim for a method that can generalize to unseen image pairs and work with a varying number of point matches and a variety of fractions of outliers.

Those goals are achieved by employing a permutation-equivariant network architecture with the following key properties: (1) a two-stage noise-aware training scheme, (2) a noise head for predicting positional inlier noise, and (3) a classification head to discriminate between inliers and outliers. These three key properties are at the core of our method. Hence, we refer to our network as a Noise-Aware Consensus Network (NACNet).

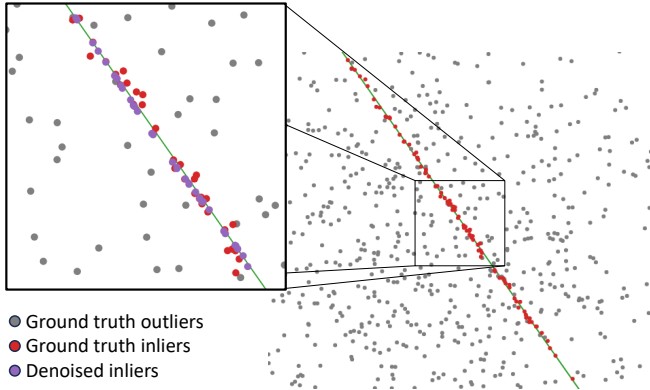

Figure 2: **NACNet point location denoising on a line-fitting task.** The set $X$ (right panel) is composed of 90% outliers (marked in grey) and (noisy) inliers (red). Our model predicts the denoised version $\hat{X}$ (purple, left panel). Evidently, the prediction of the positional noise, yielding noise-free inliers, agrees with the line model.

Formally, let $X = \{\mathbf{x}_1, \ldots, \mathbf{x}_n\} \subset \mathbb{R}^4$ denote a set of point correspondences, with $\mathbf{x}_i = (p_i, q_i, \tilde{p}_i, \tilde{q}_i)$ denoting a match between a keypoint $\mathbf{p}_i = (p_i, q_i)$ in the left image and a keypoint $\tilde{\mathbf{p}}_i = (\tilde{p}_i, \tilde{q}_i)$ in the right image. Our aim is, given $X$ as an input to classify each matching pair $\mathbf{x}_i$ as an inlier ($y_i = 1$) or outlier ($y_i = 0$) and associate with it an (inlier) confidence score $C_i \in \mathbb{R}$. Those predictions are used to estimate the essential matrix that relates the two views.

**Network architecture.** Our network comprises of three noise-aware consensus (NAC) blocks. Each NAC block uses a set encoder to map the input matches to a latent representation and to correct their positions due to positional noise. The third block further classifies the points as inliers or outliers and produces their corresponding confidence scores. Its outputs feed the model regression block, which implements a weighted differentiable Direct Linear Transformation (DLT) algorithm [34], based on the confidence scores, to predict the essential matrix. We refer the reader to Figure 1 for a detailed scheme of our network architecture.

**Noise aware consensus (NAC) block.** Each NAC block comprises of a set encoder, a noise head, and a classification head. The set encoder uses DeepSets layers to map the coordinates of the input point matches to a latent representation $L \in \mathbb{R}^{n \times d}$. Each DeepSets layer includes a linear, permutation equivariant layer followed by SoftPlus activation. These layers apply a linear transformation to each set member and an additional (different) linear transformation to their average. (We replace the sum in [55] with an average to maintain invariance to set cardinality.)

The noise and classification heads are implemented with simple two-layer MLPs. The noise head uses the latent representation to predict displacement vectors for all input points, $\delta \in \mathbb{R}^{n \times 4}$. These displacement vectors are subtracted from the input points, $X$, producing their predicted denoised locations, $\hat{X}$. The classification head uses as input the latent representation and outputs predictions for the inlier/outlier classification labels $\hat{Y} \in [0, 1]^n$ with their corresponding weights $W \in \mathbb{R}^n$.

The predicted denoised version of the keypoints $\hat{X}$ and the latent representation $L$ are passed to the next NAC block. In the third block, the inlier/outlier predicted labels $\hat{Y}$, the weights $W$, and the denoised keypoints $\hat{X}$ are passed to the model regression block.

We demonstrate the NAC block denoising effect on a simple line-fitting task. We randomly sample 100 noisy points on a line and, in addition, 900 outliers. An example is shown in Figure 2, where our NACNet significantly reduces the positional noise in the inlier points.

**Model regression block.** The model regression block uses the predicted weights, $W$, and the classification labels, $\hat{Y}$, obtained from the classification head, and the denoised version of the keypoints, $\hat{X}$, obtained from the noise head to predict the essential matrix in the following way.

$$\hat{E} = g(\hat{X}, \hat{Y}, W), \tag{1}$$

where $g$ denotes the differentiable DLT algorithm (also called the weighted eight-point algorithm, see formulation in [34], Section 3). Similarly to [46], we calculate confidence scores as follows

$$C_i = \frac{\hat{Y}_i \cdot \exp(W_i)}{\sum_j \hat{Y}_j \cdot \exp(W_j)}. \tag{2}$$

The confidence scores are used as the weights corresponding to the denoised keypoints $\hat{X}$ in the weighted DLT algorithm.

## 3.1 Loss function

We minimize a loss composed of three terms

$$L(\hat{X}, \hat{Y}, \hat{E}; X, Y, E) = L_{\text{cls}}(\hat{Y}, Y) + \alpha_{\text{mod}} L_{\text{mod}}(\hat{E}, E) + \alpha_{\text{ns}} L_{\text{ns}}(\hat{X}, X, E). \tag{3}$$

The first term $L_{\text{cls}}$ uses a weighted binary cross entropy loss, due to the imbalanced of the inliers and outliers in the data, to penalize for inlier/outlier classification errors

$$L_{\text{cls}}(\hat{Y}, Y) = -\frac{1}{n} \sum_{i=1}^{n} \left[ \beta_{\text{inliers}} \cdot y_i \cdot \log(\hat{y}_i) + \beta_{\text{outliers}} \cdot (1 - y_i) \cdot \log(1 - \hat{y}_i) \right]. \tag{4}$$

Here, $n$ is the cardinality of the keypoint set, $X$, and $\beta_{\text{inliers}}$ and $\beta_{\text{outliers}}$ are determined by a hyperparameter search.

The second term $L_{\text{mod}}$ penalizes for errors in the predicted essential matrix, similarly to the suggestion in [40]. We start with a grid of $k$ point pairs and use the Optimal Triangulation Method (OTM) [17] (page 318) to find the closest points that satisfy the ground truth epipolar constraints. Specifically, given a pair of grid points $(\mathbf{p}_i, \tilde{\mathbf{p}}_i)$, OTM seeks to find the global minimum for the following optimization problem:

$$(\mathbf{q}_i, \tilde{\mathbf{q}}_i) = \underset{\mathbf{p}'_i, \tilde{\mathbf{p}}'_i}{\operatorname{argmin}} \, d(\mathbf{p}_i, \mathbf{p}'_i)^2 + d(\tilde{\mathbf{p}}_i, \tilde{\mathbf{p}}'_i)^2 \quad \text{subject to } \tilde{\mathbf{p}}'^T_i E \mathbf{p}'_i = 0, \tag{5}$$

where $E$ is the ground truth essential matrix, and $d$ is the Euclidean distance between the points. Given the optimal points $\{\mathbf{q}_i, \tilde{\mathbf{q}}_i\}_{i=1}^{k}$, we define the loss using the Symmetric Epipolar Distance:

$$L_{\text{mod}}(\hat{E}, E) = \sum_{i=1}^{k} (\tilde{\mathbf{q}}_i^T \hat{E} \mathbf{q}_i)^2 \left( \frac{1}{\|\hat{E}^T \tilde{\mathbf{q}}_i\|_2^2} + \frac{1}{\|\hat{E} \mathbf{q}_i\|_2^2} \right), \tag{6}$$

where $\hat{E}$ is the predicted essential matrix and $k = 400$.

The last term $L_{\text{ns}}$ is used to minimize the distance between the noise-free keypoints, $\bar{X}$, and the predicted denoised version of the keypoints, $\hat{X}$, over the set of the ground truth inliers, as follows

$$L_{\text{ns}}(\hat{X}, X, E) = \|\hat{X}_{\text{inliers}} - \bar{X}_{\text{inliers}}\|. \tag{7}$$

To determine the ground truth, noise-free inlier keypoints, $\bar{X}_{\text{inliers}}$, we apply the Optimal Triangulation Method ([17], page 318). The parameters $\alpha_{\text{mod}}$ and $\alpha_{\text{ns}}$ are determined by a hyperparameter search.

## 3.2 Training

Training our model to remove outlier matches is complicated by the presence of noise in the positions of inlier matches, potentially resulting in a small classification margin. This, in turn, has been shown (in the case of kernel SVM) to have a negative effect on sample complexity and generalization error [44] (Pages 205-206, 221). A further complication is the lack of ground truth labels; i.e., our inlier/outlier labels are set by applying a preset threshold to the deviation of the points from the projections derived by the Optimal Triangulation Method ([17], page 318)(see Section 4.1).

To approach this problem, we train our model by applying a two-stage, noise-aware optimization process. The input to the first stage includes the set $\bar{X}$ containing the noise-free inlier matches along with the outlier matches. The optimization in this stage, therefore, uses only the first two terms of the loss (3), and the noise head is muted. In the second stage, the input to the network includes the original set of keypoints $X$, and the full loss, i.e., including (7), is optimized. Our experiments and ablations indicate that this two-stage training process significantly improves the performance of our method.

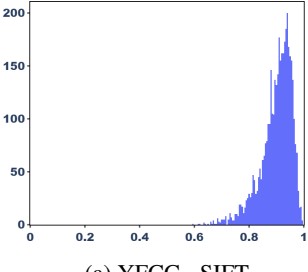 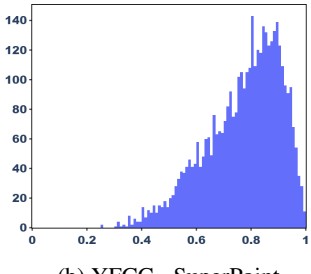 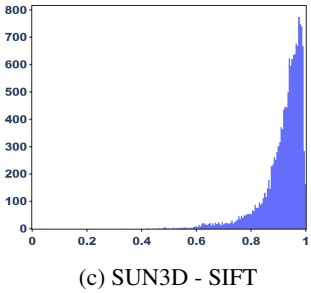

(a) YFCC - SIFT          (b) YFCC - SuperPoint          (c) SUN3D - SIFT

Figure 3: **Distributions of outliers in the different datatsets**. Histograms showing for each dataset (YFCC and Sun3D) and feature descriptor (SIFT or SuperPoint) the number of image pairs (the Y-axis) with a given fraction of outlier matches (the X-axis). The means and standard deviations (from left to right) are $0.89 \pm 0.06, 0.77 \pm 0.14, 0.92 \pm 0.08$

## 4    Experiments

### 4.1    Datasets and baselines

**Datasets.** We trained and tested our method on both indoor and outdoor datasets. For an outdoor dataset, we used Yahoo's YFCC dataset[49], which contains 100 million images from flicker later reconstructed using SFM[18]. For an indoor dataset, we used the SUN3D [53]. For both datasets, we used the same preprocessing and dataset split as in [56], i.e., the camera poses are extracted from an SFM pipeline, and the test set is split into in-scene and cross-scene generalization. In contrast to previous methods that use the Symmetric Epipolar Distance for "ground truth" inlier/outlier labeling, we determined the labels by the deviation of the points from the projections derived by the Optimal Triangulation Method ([17], page 318) using a threshold of $3 \times 10^{-3}$. In practice, changing the labeling paradigm did not affect the results. Additionally, we used the Phototourism dataset[19] to test our model's generalization across datasets. For keypoint detection, we used SIFT [30], ORB[36], and SuperPoint [11] followed by the preprocessing steps suggested in [58]. As is shown in Figure 3, consensus learning on these datasets is highly challenging due to the high fraction of outliers in all datasets and with all descriptors.

**Baselines.** We compare our methods with RANSAC[16], DEGENSAC[9], GC-RANSAC[2], MAGSAC++[4], PointNet++[38], DFE[40], LFGC[34], OA-Net[56], ACNe [32], LMC-Net[29], CL-Net[58], MS$^2$DG-Net[10], ConvMatch[57], U-Match[25], NCMNet[28], MGNet[31], BCLNet[52], and SuperGlue[42]. All the evaluations of deep learning-based methods are taken from their respective papers unless specifically stated otherwise. We used the official SuperGlue repository for evaluation on SuperPoint, and the paper[42] results for evaluation on SIFT. For the RANSAC-based methods [2, 4, 9, 16], we set the maximal number of iterations to 100K and use Lowe's ratio test[30] to filter the initial matches, with a threshold tuned differently for the SIFT and SuperPoint descriptors to maximize performance.

**Evaluation metrics.** We use the mean average precision (mAP) to evaluate our model predictions as suggested in [34]. We compute the mAP over the maximum between the translation and rotation angular errors of our predicted essential matrix up to the threshold of $5°$.

### 4.2    Essential matrix estimation

Our results are shown in Table 1-3. (Additional evaluations are shown in the Appendix.) The results demonstrate that our model outperforms the current SOTA in almost all conditions, including with indoor (SUN3D data) and outdoor (YFCC) images, with keypoint matches obtained with SIFT and SuperPoint, in in-scene (unseen image pairs from scenes included in training), cross-scene, and even cross-dataset (PhotoTourism) experiments. Specifically, in the YFCC/SIFT task (Table 1), our model outperforms the other methods by a significant margin in the in-scene generalization task and with a smaller margin in the cross-scene generalization task. Likewise, on the SUN3D dataset, our method outperforms the other methods in both in-scene and cross-scene generalization, improving over the

previous SOTA by 3.6% in the cross-scene test. Qualitative results can be seen in Figure 4 and in Appendix A.3.

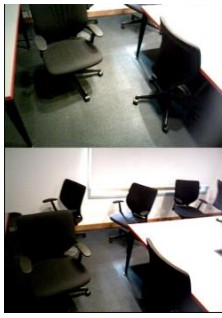 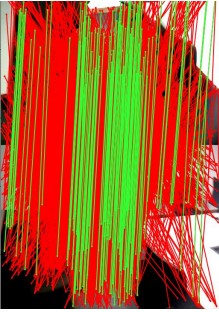 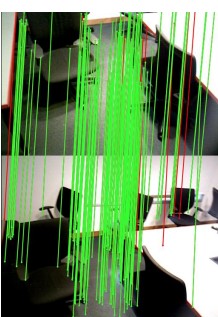

Figure 4: **NACNet inlier/outlier classification**. An example from the SUN3D dataset. Left to right: input image pairs, input matches, and our model's predicted inliers. Color mark ground truth labels: inlier matches are marked in green; outliers are marked in red.

Table 1: **SIFT evaluation**. Evaluation of essential matrix estimation on the YFCC and SUN3D datasets with keypoint matching obtained with SIFT. mAP5°(%) is reported, and the best result in each column is in bold. In-scene denote results on novel image pairs taken from scenes that were included in the training data and cross-scene denote results on image pairs taken from unseen scenes. The first set of methods (above the middle line) includes methods that incorporate RANSAC.

| Method | YFCC(%) | | SUN3D(%) | |
|---|---|---|---|---|
| | In-scene | Cross-scene | In-scene | Cross-scene |
| RANSAC | 31.57 | 42.78 | 20.88 | 15.79 |
| GC-RANSAC | 30.88 | 42.55 | 18.69 | 13.57 |
| LO-RANSAC | 30.96 | 42.60 | 19.01 | 13.85 |
| MAGSAC++ | 31.01 | 42.57 | 19.55 | 14.23 |
| SuperGlue | - | 59.25 | - | - |
| Point-Net++ | 10.49 | 16.48 | 10.58 | 8.10 |
| DFE | 19.13 | 30.27 | 14.05 | 12.06 |
| LFGC | 13.81 | 23.95 | 11.55 | 9.30 |
| OA-Net++ | 32.57 | 38.95 | 20.86 | 16.18 |
| ACNe | 29.17 | 33.06 | 18.86 | 14.12 |
| LMC-Net | 33.73 | 47.50 | 19.92 | 16.82 |
| CL-Net | 39.16 | 53.10 | 20.35 | 17.03 |
| MS$^2$DG-Net | 38.36 | 49.13 | 22.20 | 17.84 |
| ConvMatch | 43.48 | 54.62 | 25.36 | 21.71 |
| U-Match | 46.78 | 60.53 | 24.98 | 21.38 |
| NCMNet | 52.33 | 63.43 | 26.12 | 20.66 |
| MGNet | 51.43 | 64.63 | 25.96 | 21.27 |
| BCLNet | 52.62 | 66.08 | 24.59 | 19.96 |
| Ours | **60.10** | **66.14** | **34.15** | **25.36** |

We also tested our model using the deep-learning based descriptor SuperPoint on both the YFCC and SUN3D datasets. Table 2 shows that our model improves over existing SOTA methods that do not incorporate RANSAC by 6.9% on the cross-scene YFCC test and by 2.96% on the cross-scene SUN3D test. Overall, our model achieves SOTA results over both datasets and descriptors.

## 4.3 Generalization to different descriptors and datasets

We next examined our model's ability to generalize across different descriptors and datasets. For these experiments we used our model trained on the YFCC dataset with SIFT matches. We then tested this model on image pairs from novel scenes (i.e., cross-scene experiment) from the YFCC dataset with matches obtained using ORB and SuperPoint. Additionally, we applied the YFCC/SIFT model to image pairs from the test set of the Phototurisem dataset with matches obtained with SIFT and SuperPoint. The results are shown in Table 3. While our model slightly underperforms the SOTA with ORB matches, it outperforms existing methods with SP matches by almost 3.9%. On

Table 2: **SuperPoint evaluation**. Evaluation of camera pose estimation in experiments on outdoor and indoor datasets using matches obtained with SuperPoint. mAP5°(%) is reported, and the best result in each column is in bold. In-scene denote results on novel image pairs taken from scenes that were included in the training data and cross-scene denote results on image pairs taken from unseen scenes. The first set of methods (above the middle line) includes methods that incorporate RANSAC.

| Method | YFCC(%) | | SUN3D(%) | |
|---|---|---|---|---|
| | In-scene | Cross-scene | In-scene | Cross-scene |
| RANSAC | 20.36 | 25.60 | 21.25 | 15.89 |
| GC-RANSAC | 17.27 | 22.27 | 19.32 | 14.17 |
| LO-RANSAC | 17.25 | 22.12 | 19.22 | 14.46 |
| MAGSAC++ | 18.09 | 23.47 | 20.23 | 15.02 |
| SuperGlue | 39.71 | 57.45 | 24.09 | 19.45 |
| Point-Net++ | 11.87 | 17.95 | 11.40 | 9.38 |
| DFE | 18.79 | 29.13 | 13.35 | 12.04 |
| LFGC | 12.18 | 24.25 | 12.63 | 10.68 |
| OA-Net++ | 29.52 | 35.27 | 20.01 | 15.62 |
| ACNe | 26.72 | 32.98 | 18.35 | 13.82 |
| CL-Net | 29.35 | 38.99 | 15.89 | 14.03 |
| MS$^2$DG-Net | 30.40 | 37.38 | 20.28 | 16.08 |
| U-Match | 35.12 | 45.72 | 22.73 | 18.87 |
| ConvMatch | 38.34 | 48.80 | 25.36 | 21.71 |
| NCMNet | – | 52.20 | – | – |
| MGNet | 41.53 | 49.37 | 24.58 | 20.65 |
| BCLNet | 40.56 | 48.07 | - | - |
| Ours | **55.94** | **59.10** | **33.97** | **24.67** |

Table 3: **Generalization across descriptors and datasets**. This table shows the results obtained with our model trained on the YFCC dataset with SIFT matches applied in inference to the YFCC dataset with the ORB and SuperPoint (SP) feature extractors and to the PhotoTourism dataset with SIFT and SuperPoint (SP). mAP5°(%) is reported, and the best result in each column is marked in bold. † indicates evaluation conducted using published models.

| | YFCC(%) | | PhotoTourism(%) | |
|---|---|---|---|---|
| | ORB | SP | SIFT | SP |
| LFGC | 7.40 | 14.78 | 20.17 | 5.89 |
| OA-Net++ | 12.05 | 19.40 | 40.39 | 8.99 |
| CL-Net | 14.75 | 21.00 | 45.54 | 9.41 |
| MS$^2$DG-Net | 11.38 | 21.05 | 45.53 | 12.91 |
| U-Match | 16.70 | 28.38 | 54.43 | 11.48 |
| NCMNet | 19.95 | 33.20 | 54.73 | 30.60 |
| BCLNet$^†$ | 18.70 | 25.85 | 54.29 | 23.34 |
| MGNet | **20.00** | 32.88 | 57.64 | 20.41 |
| Ours | 19.17 | **37.14** | **60.81** | **49.03** |

generalization to the Phototurism dataset our method performs best with both SIFT and SuperPoint matches.

## 4.4 Resource utilization

In Table 4, we provide an account of the resources used by our model including the number of parameters, GPU memory usage, and runtime. It can be seen that while our model uses more parameters than NCMNet and BCLNet, which use graph attention architectures, it is 4-6 times faster than these methods and consumes less GPU memory at inference. We further compare our runtime to RANSAC-based methods. With 100K iterations, RANSAC is significantly slower than our method. We note that RANSAC is implemented in CPU. We further considered the recent Kornia's GPU implementation of RANSAC for fundamental matrix estimation [41]. Using a batch size of 10000 samples, their model runtime and maximum GPU memory usage were 40.94ms and 414.66MB, respectively, which are 4 times higher than our model. For GPU resource usage, we used an NVIDIA GeForce RTX 2080Ti and an Intel(R) Xeon(R) Gold 6248 CPU @ 2.50GHz for CPU.

Table 4: **Resource utilization.** The table shows the resource usage of our model compared to previous methods in terms of number of parameters, GPU memory usage, and runtime. Tested on YFCC with SIFT descriptors. Note that GC-RANSAC and MAGASAC++ are implemented in CPU.

| Methods | #Params (M) | Max GPU Mem (MB) | Runtime Avg.(ms) |
|---|---|---|---|
| GC-RANSAC | - | - | 217.3 |
| MAGASAC++ | - | - | 295.29 |
| NCMNet | **4.77** | 174.52 | 67.43 |
| BCLNet | 4.87 | 140.91 | 46.94 |
| Ours | 22.14 | **130.75** | **11.12** |

## 4.5 Ablation study

**Keypoint denoising.** In ablation studies we tested the importance of our noise head, i.e., keypoint denoising process, by training our model with and without this head. In both cases, we used our two stage training scheme. The results in Table 5 demonstrate that our keypoint denoising improves our model performance in both indoor (SUN3D) and outdoor (YFCC) scenes and using both SIFT and SuperPoint matches. This improvement is more noticeable in the more challenging indoor scenario, as it includes fewer inliers and less accurate positions of keypoints.

**Two-stage noise-aware training.** To test the importance of our two-stage training scheme, we trained our model in a single stage on the original (noisy) set of keypoints $X$, while the noise head is muted on the YFCC dataset with SIFT matches. The results are shown in Table 5 (first row vs. second row). The model trained in a single stage performs similarly to the two-stage trained model in the in-scene generalization test and significantly worse in the cross-scene test.

**Correspondences pruning.** Previous works [58, 28, 52] used correspondence pruning to reduce the effect of the outliers' distribution on the final prediction. Specifically in these schemes, matches with the lowest classification scores were removed after each block in their networks. To test the effect of iterative pruning in our model, we implemented a similar scheme, removing half of the input matches with the lowest scores after each block. We test this pruning strategy on a model trained on YFCC with SIFT matches without keypoint denoising. In contrast to results reported for previous methods, the pruning process had a slightly negative effect on our model prediction, decreasing its mAP5° cross-scene score from 65.32% to 64.52%, probably because pruning may also remove some inlier matches. This experiment suggests that our NAC blocks can handle large numbers of outliers successfully without the need for additional pruning.

Table 5: **Ablation studies**. Evaluation of our model without keypoint denoising and 2-stage training. mAP5°(%) is reported, and the best result in each column and dataset/descriptors is marked in bold. In-scene denote results on novel image pairs taken from scenes that were included in the training data and cross-scene denote results on image pairs taken from unseen scenes.

| Dataset | Descriptor | 2-stage training | Keypoint denoising | In-scene | Cross-scene |
|---|---|---|---|---|---|
| YFCC | SIFT | – | – | 57.31 | 59.70 |
| | | ✓ | – | 58.95 | 65.32 |
| | | ✓ | ✓ | **60.10** | **66.14** |
| | SuperPoint | ✓ | – | 54.65 | 58.64 |
| | | ✓ | ✓ | **55.94** | **59.10** |
| SUN3D | SIFT | ✓ | – | 30.73 | 23.02 |
| | | ✓ | ✓ | **34.15** | **25.36** |
| | SuperPoint | ✓ | – | 27.19 | 21.67 |
| | | ✓ | ✓ | **33.97** | **24.67** |

**Hyper-parameter search.** We tested our model with different hyper-parameters, including the number of NAC blocks (which affects the number of DeepSet layers) and the encoder dimension. Using two NAC blocks instead of three reduces the mAP5° to 53.52% and 59.75% for in-scene and

cross-scene, respectively. Setting the encoder dimension to 256 instead of 512 reduces the mAP5° to 54.86% and 63.47%. These hyper-parameters search further justify our choices.

### 4.6 Implementation details

**Training.** At first, we trained our model on the YFCC dataset with SIFT matches (25 epochs in the noise-free pretraining stage and an additional 10 epochs in the second stage). Then, to save on resources, we finetuned the noise-free pretrained YFCC/SIFT model to initialize the training of the rest of the models (SUN3D and YFCC with SuperPoint features), for another 5 epochs with the respective noise-free dataset. Lastly, we train these pretrained models for 10 more epochs using the original (noisy) data. Training was run on an NVIDIA Quadro RTX 6000/ DGX V100/ A40 GPUs, with a maximum memory usage of 5GB.

For the loss function we set $\beta_{\text{inliers}} = 1, \beta_{\text{outliers}} = 10, \alpha_{\text{mod}} = 1$ and $\alpha_{\text{ns}} = 100$. We used a threshold of $3 \times 10^{-3}$ for labeling inliers and outliers. In training, we used the ADAM[23] optimizer with a batch size of 32 image pairs and a learning rate of $10^{-4}$. We note that in the noise-free pretraining stage, the predictions of all three blocks are considered in the loss function, whereas in the second stage, only the prediction of the third block is considered.

**Architecture details.** The network consists of three consecutive NAC blocks, where we only use the output of the last block at inference time. The Set Encoders in the NAC blocks combine 12 set layers interleaved with SoftPlus activation, layer normalization, and skip connections in a ResNet-like architecture. We set the dimension of the Set Encoder to 512. The Classification and Noise Heads consist of two-layer MLPs interleaved with an activation function. We used a SoftPlus activation for the classification head and a LeakyReLU for the noise head. The classification head produces an $n \times 2$ vector. We apply a sigmoid function on the first coordinate to predict $\hat{Y}$ and use the second coordinate for the weight prediction $W$.

## 5   Conclusion

We presented NACNet, a Noise-Aware Deep Sets framework to estimate relative camera pose, given a set of putative matches extracted from two views of a scene. We demonstrated that a position denoising of inliers and noise-free pretraining enable accurate estimation of the essential matrix. Our experiments indicate that our method can handle large numbers of outliers and achieve accurate pose estimation superior to previous methods. We generally observed good cross-dataset and cross-descriptor generalization compared to existing methods, but hope to further improve on those in future work. In addition, we believe adding a block performing degeneracy test, can further help properly utilizing non-degenerate configurations of matches and consequently improve the results of the DLT block. Finally, in future work, we will seek to incorporate our work in multiview structure from motion pipelines.

## Acknowledgments and Disclosure of Funding

This research was supported in part by the Israel Science Foundation, grant No. 1639/19, by the Israeli Council for Higher Education (CHE) via the Weizmann Data Science Research Center, by the MBZUAI-WIS Joint Program for Artificial Intelligence Research and by research grants from the Estates of Bernice Bernath and Marni Josephs Grossman; Joel B. Levey; Tully and Michele Plesser and the Anita James Rosen and Harry Schutzman Foundations.

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

# A  Appendix

Below, we include further quantitative evaluations, a comparison of visual results between our method and previous methods, and a list of used assets' licenses.

## A.1  Denoising evaluation

Below we evaluate the utility of the denoising component of our network. Figure 5 shows the noise distribution before and after our keypoint denoising, measured according to the ground truth essential matrix. The median of the mean reprojection error (over each pair) reduces due to this component by 0.202 pixels and even more (0.246 pixels) for image pairs with pose error (maximum between the translation and rotation errors) lower than five degrees.

Figure 5: **Denoising evaluation.** Reprojection error of inlier keypoints before and after applying our denoising scheme, computed using the ground truth pose. The box plots show the 0.25, 0.5, and 0.75 quantiles. The two left bar plots represent the evaluation over all the image pairs in the YFCC dataset. The right two bar plots focus on image pairs whose pose prediction was accurate (i.e., pose error below $5°$, where the pose error is defined as the maximum of the translation and rotation angular errors.). Evaluation was conducted on the YFCC dataset using SIFT descriptors.

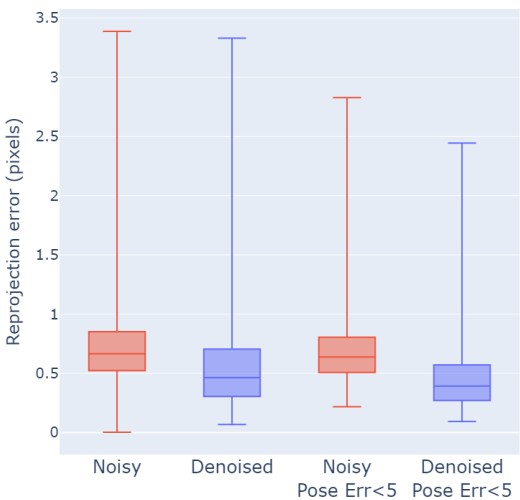

## A.2  Classification evaluation

Our inlier/outlier classification results are shown in Table 6. "Ground truth" labels are determined by triangulation, as discussed in Section 3.2. It can be seen that our model achieves higher F1 scores compared to previous methods, possibly explaining our overall improved regression accuracy.

## A.3  Qualitative results

Figure 6 shows results obtained with our method, compared with CLNet and MGNet. Here we use the same indoor and outdoor image pairs shown in the MGNet paper. In Figure 7 we compare our results to NCMNet (using their published checkpoint) on randomly selected image pairs from the YFCC dataset. It can be seen in both figures that our method generally outputs fewer outliers than previous methods.

## A.4  License of used assets

- YFCC100M dataset[49] images are under a common-creative license, and each media file in the dataset is subject to the Creative Commons licenses chosen by their creators/uploaders.
- SUN3D dataset[53] is published under MIT license.

Table 6: **Classification evaluation.** Inlier/outlier classification results on the YFCC dataset and SIFT descriptors. Precision(P), Recall(R), and F1 score are reported.

| Methods | In-scene | | | Cross-scene | | |
|---|---|---|---|---|---|---|
| | $P(\%)$ | $R(\%)$ | $F1(\%)$ | $P(\%)$ | $R(\%)$ | $F1(\%)$ |
| RANSAC | 47.4 | 52.6 | 49.9 | 43.5 | 50.6 | 46.8 |
| PointNet++ | 49.8 | 86.4 | 63.2 | 46.6 | 84.1 | 59.9 |
| LFGC | 56.6 | 86.3 | 68.3 | 54.6 | 84.7 | 66.4 |
| OANet++ | 60.0 | 89.3 | 71.8 | 55.7 | 85.9 | 67.6 |
| MSA-Net | 61.9 | 90.5 | 73.5 | 58.7 | 87.9 | 70.4 |
| CLNet | 76.0 | 79.2 | 77.6 | 75.0 | 76.4 | 75.7 |
| MS$^2$DG-Net | 63.1 | 90.9 | 74.5 | 59.1 | 88.4 | 70.8 |
| ConvMatch | 63.0 | **91.5** | 74.6 | 58.7 | **89.3** | 70.9 |
| NCMNet | 78.4 | 81.7 | 79.6 | 77.0 | 78.2 | 77.4 |
| BCLNet | 78.4 | 82.5 | 80.1 | 77.3 | 79.7 | 78.3 |
| Ours | **84.6** | 82.9 | **83.2** | **82.2** | 79.1 | **80.2** |

- Phototourism dataset[19] is published under Creative Commons Attribution-NonCommercial-ShareAlike 4.0 International License: CC BY-NC-SA 4.0

- SuperGlue[42] code and weights are published under a license for:"ACADEMIC OR NON-PROFIT ORGANIZATION NONCOMMERCIAL RESEARCH USE ONLY"

- SuperPoint[11] code and weights are published under a license for:"ACADEMIC OR NON-PROFIT ORGANIZATION NONCOMMERCIAL RESEARCH USE ONLY"

- part of our code is adopted from CLNet[58] which is published under GPL-3.0 license.

- BCLNet[52] and NCMNet[28] code is published without specifying a license.

- OPENCV[6] and Kornia[41] are published under Apache License 2.0

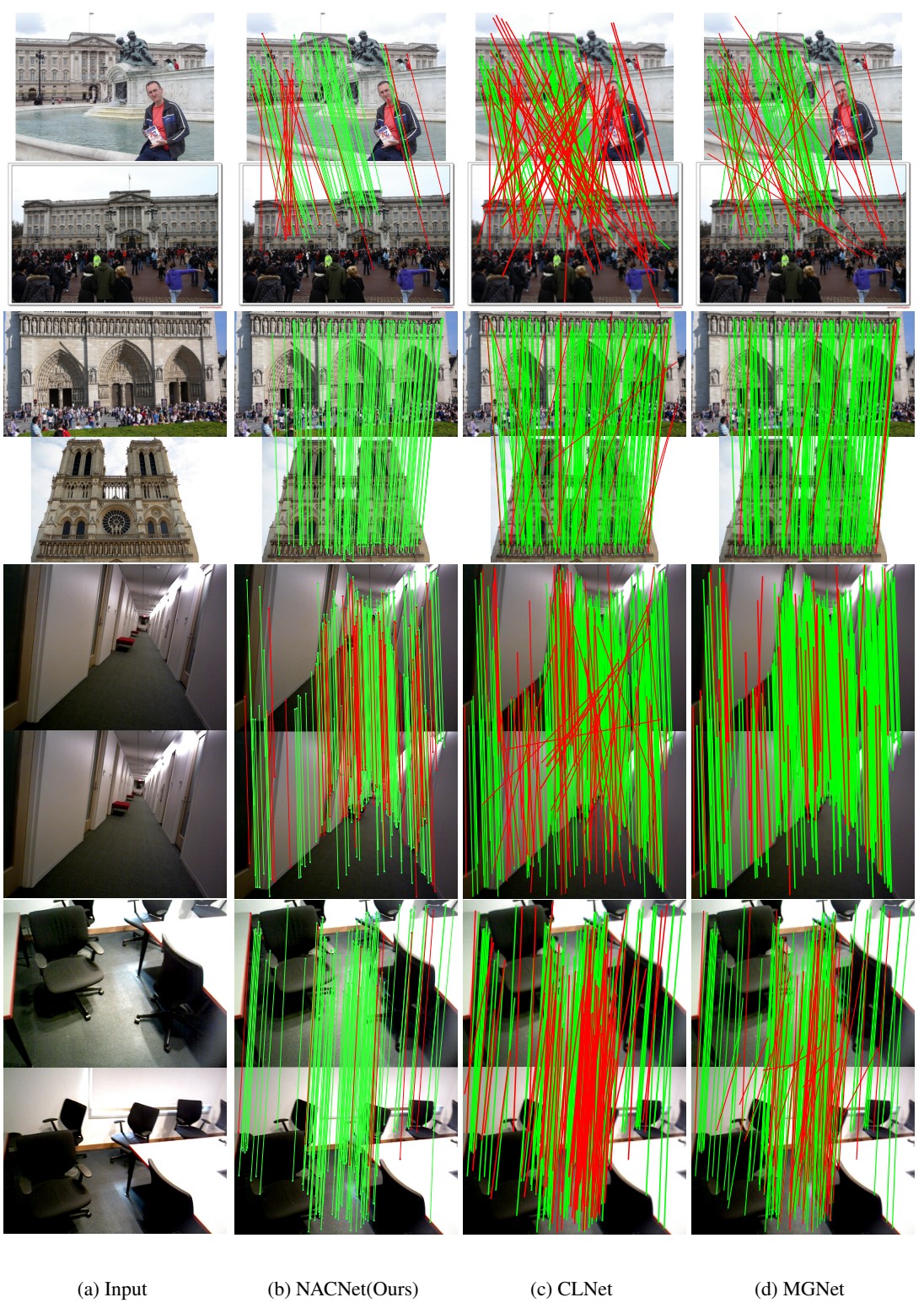

|                |                   |           |           |
|:--------------:|:-----------------:|:---------:|:---------:|
| (a) Input      | (b) NACNet(Ours)  | (c) CLNet | (d) MGNet |

Figure 6: **Qualitative results**. Visualization results of two-view correspondence pruning on unknown outdoor and indoor scenes. From left to right are the input pairs and the results of NACNet, CLNet, and MGNet, respectively. Inliers are marked in green and outliers are marked in red.

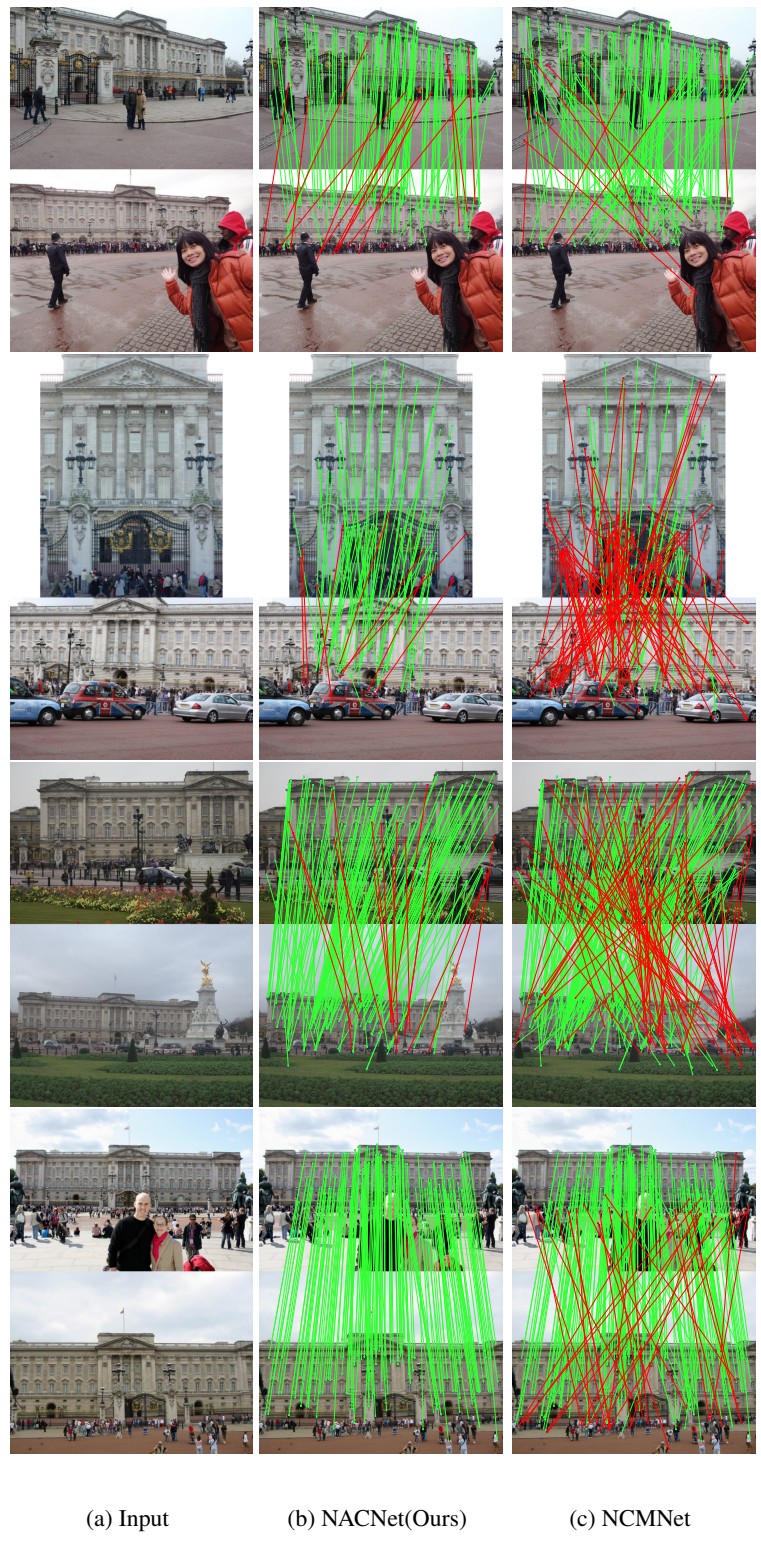

       (a) Input          (b) NACNet(Ours)       (c) NCMNet

Figure 7: **Qualitative results**. Visualization results of two-view correspondence pruning on unknown outdoor scenes. From left to right are the input pairs and the results of NACNet and NCMNet, respectively. Inliers are marked in green and outliers are marked in red.

