# OpenReview forum: "Consensus Learning with Deep Sets for Essential Matrix Estimation"
_NeurIPS.cc/2024/Conference — NeurIPS 2024 poster_

### Official Review · Reviewer_1GwE · 2024-07-11

**Soundness:** 2
**Presentation:** 3
**Contribution:** 3
**Rating:** 6
**Confidence:** 5

**Summary:**

This work proposes a simpler yet effective network architecture based on Deep Sets for the estimation of essential matrices.

The method is based on three key properties:

1. A two-stage noise-aware training scheme first uses noise-free inlier matches for training, and then uses original matches and full loss for training;

2. A noise head predicts the displacement noise in inlier matches.

3. A classification head classifies point matches as either inliers or outliers.

Evaluations show that the method outperforms previous methods in terms of accuracy and generalization.

**Strengths:**

1. The proposed network architecture is simple yet effective.

2. NAC block represents denoising and inlier/outlier classification effects, well validated through a toy example, i.e., a line-fitting task in Fig. 2.

3. The key properties proposed are shown to be effective in the ablation study.

4. The paper is well-structured and is generally easy to follow.

**Weaknesses:**

1. While the DeepSets-based architecture is shown to be effective, traditional fully convolutional networks, such as UNet, are also applicable. Is there any strong reason why a DeepSets-based network is preferred and explored first?

2. The authors remove half of the input matches to test the effect of outlier pruning. Is this approach reasonable? Is this value adjustable? It would be better to add an ablation study showing performance changes as the number of removed matches varies. Furthermore, this is an indirect method to validate the effectiveness of the proposed inlier/outlier classification head. Why was the ablation study not conducted with and without this head?

3. While the paper provides an ablation study on the proposed key properties, it lacks an ablation of the network architecture and hyperparameters, such as the number of set layers, the dimension of the Set Encoder, etc, but this would also be valuable to know — for example, how much of an impact do these parts of the network have on the final results?

4. The paper lacks details on applying the Optimal Triangulation Method to set inlier/outlier labels and obtain ground truth, noise-free inlier keypoints. Including these details would help readers better understand the training process.

5. What is the unit of the y-axis in Fig. 3?

6. To the best of my knowledge, the ground truth poses in the SUN3D dataset are inaccurate (see supplemental material in {1}). Why did the authors use this dataset solely to follow the evaluation protocol of previous methods? For indoor scenes, is it more suitable to use ScanNet dataset {2} for evaluation as [42]?

7. While the paper claims that the network is simpler than existing methods, the multiple NAC blocks and the noise regression module may introduce a certain level of complexity. It would be better to provide experiments and discussions on the computational efficiency of the proposed method compared to existing methods, including runtime and model parameter scale, which are critical for real-world applications.

{1} Bellavia, Fabio. “Image Matching by Bare Homography.” IEEE Transactions on Image Processing 33 (2023): 696-708.
{2} Dai, Angela, Angel X. Chang, Manolis Savva, Maciej Halber, Thomas A. Funkhouser and Matthias Nießner. “ScanNet: Richly-Annotated 3D Reconstructions of Indoor Scenes.” IEEE Conference on Computer Vision and Pattern Recognition (CVPR) (2017): 2432-2443.

**Questions:**

1. While the DeepSets-based architecture is shown to be effective, traditional fully convolutional networks, such as UNet, are also applicable. Is there any strong reason why a DeepSets-based network is preferred and explored first?

2. The authors remove half of the input matches to test the effect of outlier pruning. Is this approach reasonable? Is this value adjustable? It would be better to add an ablation study showing performance changes as the number of removed matches varies. Furthermore, this is an indirect method to validate the effectiveness of the proposed inlier/outlier classification head. Why was the ablation study not conducted with and without this head?

3. While the paper provides an ablation study on the proposed key properties, it lacks an ablation of the network architecture and hyperparameters, such as the number of set layers, the dimension of the Set Encoder, etc, but this would also be valuable to know — for example, how much of an impact do these parts of the network have on the final results?

4. The paper lacks details on applying the Optimal Triangulation Method to set inlier/outlier labels and obtain ground truth, noise-free inlier keypoints. Including these details would help readers better understand the training process.

5. What is the unit of the y-axis in Fig. 3?

6. To the best of my knowledge, the ground truth poses in the SUN3D dataset are inaccurate (see supplemental material in {1}). Why did the authors use this dataset solely to follow the evaluation protocol of previous methods? For indoor scenes, is it more suitable to use ScanNet dataset {2} for evaluation as [42]?

7. While the paper claims that the network is simpler than existing methods, the multiple NAC blocks and the noise regression module may introduce a certain level of complexity. It would be better to provide experiments and discussions on the computational efficiency of the proposed method compared to existing methods, including runtime and model parameter scale, which are critical for real-world applications.

{1} Bellavia, Fabio. “Image Matching by Bare Homography.” IEEE Transactions on Image Processing 33 (2023): 696-708.
{2} Dai, Angela, Angel X. Chang, Manolis Savva, Maciej Halber, Thomas A. Funkhouser and Matthias Nießner. “ScanNet: Richly-Annotated 3D Reconstructions of Indoor Scenes.” IEEE Conference on Computer Vision and Pattern Recognition (CVPR) (2017): 2432-2443.

---

> ### Author Rebuttal · Authors · 2024-08-07
>
> We thank the reviewer for these comments.
>
> Q1. UNet:
>
> Our network obtains as input an unordered set of point matches of apriori unknown cardinality. Such input is naturally suitable for DeepSets and other permutation equivariant architectures. Standard convolutional and the UNet architecture cannot handle such input, since it requires input on a grid. Graph convolutional architectures can be applied (and indeed are applied in previous methods, including CLNet, NCMNet, and also U-Match, which uses a graph UNet architecture). These methods, however, are more complex and, as our experiments show, are slower (see the attached Table R.1) and yield inferior accuracies (see Tables 1-2 in the paper).
>
> Q2. Pruning amount:
>
> For our ablation, we followed previous approaches, including CLNet, NCMNet, and BCLNet, which pruned half of the input matches. MGNet further performed an ablation study with different pruning ratios and showed that pruning reduces the accuracy of the predictions. We should emphasize that our confidence scores (Eq. 2 in the paper) perform “soft pruning,” which appears to be more effective than “hard pruning.”
>
> Ablating the classification head:
>
> We evaluate the performance of our model without the classification head. Due to the prevalence of outliers, performance deteriorates to chance levels (0.014% and 0.025% mAP5 score for in-scene and cross-scene, respectively). We further demonstrate the effectiveness of our classification head by providing its precision, recall, and F1 scores (Table R.3).
>
> Q3. Hyperparameters ablation studies:
>
> We include in Table R.5 an ablation of the number of NAC blocks (which affects the number of DeepSet layers) and the encoder dimension. These ablations further justify our choices.
>
> Q4. Optimal Triangulation Method:
>
> We will add these details to the paper.
>
> Q5. Y-axis units in Fig. 3:
>
> The Y-axis represents the number of image pairs.
>
> Q6. ScanNet evaluation:
>
> Indeed, we used SUN3D to follow the evaluation protocol of previous methods. For the rebuttal, we include in Table R.2 the results of cross-dataset generalization on ScanNet. We compared our results to NCMNet predictions. Our method achieves superior results on this highly challenging dataset. Since previous methods have not published their model trained on SUN3D, we trained NCMNet from scratch using the official code release.
>
> Q7. Resources usage:
>
> We address this question in Table R.1. While our model uses more parameters than NCMNet and BCLNet, which use graph attention architectures, it is 4-6 times faster than these methods and consumes less GPU memory at inference.

---

> > ### Comment · Reviewer_1GwE · 2024-08-10
> > **Thanks for the authors' feedback.**
> >
> > I read the rebuttal and all other reviews. This rebuttal has addressed most of my concerns, but some questions still remain.
> >
> > 1. For the reply to Q6, could you please provide the results of other baseline methods on ScanNet dataset?
> >
> > 2. For the reply to Q7, somewhat counterintuitively, the model achieves a faster runtime despite using more parameters.

---

> > > ### Author Response · Authors · 2024-08-12
> > >
> > > Thank you for your response and questions.
> > >
> > >
> > > 1. Here are the results, with two more baseline methods:
> > > | Method                  | MAP5  | MAP10  | MAP20  | MAP30  |
> > > |----------------------|-------|--------|--------|--------|
> > > | U-Match              |  6.53                 | 13.16 | 22.84  | 30.44  |
> > > | ConvMatch w/RANSAC   | 6.53                 | 15.15 | 26.54  | 33.77  |
> > > | NCMNet               |8.6   | 15.16  | 25.65  | 33.56  |
> > > | NACNet (Ours)        |  **10.33** | **17.9**   | **28.55** | **36.08**  |
> > > We note that other previous methods did not provide pre-trained models for SUN3D or config files with hyperparameters.
> > >
> > > 2. Indeed, our method appears to achieve a faster runtime (11.12ms) despite using more parameters. This is due to the simplicity of our network’s architecture, which utilizes 82 standard fully-connected layers. In comparison, BCLNet (runtime 46.94), interleaves k-nearest neighbors (k-nn), graph convolution, and attention layers within a total of 190 layers. Further profiling of their code shows that the k-nn (which involves no parameters) takes 16% of this time (\~7.5ms), and the convolutions take 12% (\~5.6ms). Moreover, the forward pass over 190 layers is sequential and cannot be parallelized as efficiently as shallower networks.

---

### Official Review · Reviewer_wTxF · 2024-07-13

**Soundness:** 3
**Presentation:** 3
**Contribution:** 3
**Rating:** 6
**Confidence:** 3

**Summary:**

The paper proposes a deepsets based architecture for essential matrix estimation. The proposed architecture is required to overcome positional nosie, be capable of handling outlier matches which comprise a significant portion of the input data, generalize to unseen image pairs and be capable of handling arbitrary number of point matches.

The propose archchitecture comes with the following:

  - a two-stage noise-aware training scheme

  - a noise head for predicting positional inlier nosie, and

  - a classification head for inlier/outlier discrimination.

The architecture consists of

  - Nosie-aware Concensus Blocks (NAC)

    - These are stacks of set encoders (ie Deepsets) used to obtain latent representation of input sets and for classification of points as inliers or outliers.

  The ouput of these blocks are used to estimate the essential matrix.

**Strengths:**

- A simple architecture is proposed that offers better empirical performance compared to current methods.

**Weaknesses:**

- Ablation on the choice of set functions is missing.

**Questions:**

- Can an ablation on the choice of set functions be provided? For instance, recent methods such as [1] and [2] have been shown to perform better than deepsets for modeling interactions over sets and these might provide further performance boost to the proposed architecture.

- Also, how does the number of deepset layers affect performance? [3] shows that special techniques are required to ensure effective learning when multiple deepsets layers are stacked. Curious how this correlates with your experiments.

## References

[1] Lee, Juho, et al. "Set transformer: A framework for attention-based permutation-invariant neural networks." International conference on machine learning. PMLR, 2019.

[2] Bruno, Andreis, et al. "Mini-batch consistent slot set encoder for scalable set encoding." Advances in Neural Information Processing Systems 34 (2021): 21365-21374.

[3] Zhang, Lily, et al. "Set norm and equivariant skip connections: Putting the deep in deep sets." International Conference on Machine Learning. PMLR, 2022.

**Limitations:**

While no discussion of limitations is provided, i do not see any immediate negative societal impact of this work.

---

> ### Author Rebuttal · Authors · 2024-08-07
>
> We thank the reviewer for these comments.
>
> Q1.Set transformer ablation:
>
> We tested the SetTransformer in a quick synthetic experiment. We trained both the SetTransformer and our DeepSet model on noise-free data. While our model achieves a highly accurate pose estimation of 86.52% mAP5 (due to the lack of noise), the SetTransformer model achieves much less accurate results of 66.79% mAP5. We believe the SetTransformer architecture suffers from the lack of global features and a narrow bottleneck in the ISAB block. We leave improvement of the SetTransformer architecture to future work.
>
> Q2. Number of DeepSet layers:
>
> We provide further ablations in Table R.5, showing that the removal of one NAC block (two blocks instead of three) reduces accuracy by nearly 7%.
>
> Thank you also for bringing up the work of Zhang et. al [3]. Despite some architectural differences, our implementation respects the principle of keeping a “clean path” for the gradient descent (e.g., by utilizing skip connections in a similar design as in [3]), enabling the usage of multiple DeepSet layers.

---

### Official Review · Reviewer_HiXm · 2024-07-13

**Soundness:** 4
**Presentation:** 4
**Contribution:** 3
**Rating:** 8
**Confidence:** 4

**Summary:**

In this work, authors propose NACNet (Noise Aware Consensus network) for the robust essential matrix estimation. For this purpose authors apply DeepSets based architecture that predicts inlier / outlier class as well as inlier displacement error estimation. Authors also propose a two-stage training: (1st) train with noise-free matches by disabling noise-prediction head and (2nd) train on real data with noise.

Extensive experiments with in-scene, cross-scene, cross-dataset, cross-feature generalization and extensive ablation study shows that propose simpler method is superior compared to other deep-learning based consensus methods with complicated architectures.

**Strengths:**

Overall, paper is well written with sufficient mathematical rigor to prove authors claim that "Our network achieves accurate recovery that is superior to existing networks with significantly more complex architectures."

(1) Predicting noise in key-points without feature-descriptor is well-motivated and proven to work through ablation study.

(2) Architecture design is simple with DeepSet based 3 NAC block with Model Regression head on top. The network predicts confidence along with inlier-outlier classification (similar to [1]) that allows robust estimation of Essential Matrix via differentiable DLT.

(3) Whole network is end-to-end differentiable.

(4) Benchmarkin &, experiments across different datasets shows good generalization capability of the proposed method. This can have high impact in relative-pose estimation that is essential to multi-view geometry.

References

[1] Yi, Kwang Moo, et al. "Learning to find good correspondences." Proceedings of the IEEE conference on computer vision and pattern recognition. 2018.

**Weaknesses:**

Paper discusses predicting noise-free key-points as an output, however lacks sufficient discussion around difference between noisy and noise-free points.

(1) How does keypoint noise changes feature points? It would have been nicer to see some visualization / empirical matrix of change in key-point locations. of noise-free key-points predicted.

(2) Another interesting fact that is not tested: does the change in key-point position is to fit into the "Model" or is it actually becoming sub-pixel accurate as shown in [2]

(3)  Some more comparison when input has less "outlier" / more "outlier". Empirical evidence around effect of cardinality is also missing.


References

[2] Lindenberger, Philipp, et al. "Pixel-perfect structure-from-motion with featuremetric refinement." Proceedings of the IEEE/CVF international conference on computer vision. 2021.

**Questions:**

L97: "Our work is first one to apply keypoint position correction without using a geometric model" -- Last loss that defines distance between noise-free keypoints and prediction uses "geometry" (as in optimal triangulation) to get noise-free points. What does it mean here when authors say "noise-free"?

What is the size of the model? How fast is the inference? It would be interesting to know how close are we compared to RANSAC in-terms of runtime?

**Limitations:**

Authors discuss limitation of their approach especially not-perfect generalization to other datasets / features, adding degeneracy test block inside the model.

This work doesn't have societal impact, so authors ignore that.

---

> ### Author Rebuttal · Authors · 2024-08-07
>
> We thank the reviewer for these comments.
>
> W1.Denoising evaluation:
>
> We include a summary box plot (Figure R.1) of the noise distribution before and after our key-point denoising, measured with respect to the ground truth essential matrix. The median of the mean reprojection error (over each pair) is reduced by 0.202 pixels and even more (0.246 pixels) for image pairs with pose error (maximum between the translation and rotation errors) lower than five degrees.
>
> W2. Noise prediction:
>
> In principle, our denoising module is trained with correspondences cleaned with the ground truth essential matrix, and so it is expected to predict noise-free key-point locations independent of the predicted essential matrix. Moreover, as our ablation indicates (Table 4 in the paper), this denoising process improves the accuracy of the essential matrix regression, indicating that it yields a better fit to true key-point positions rather than to the predicted model.
>
> W3. Performance with different outlier ratios:
>
> We address this question in Table R.4. As expected, accuracy drops as the fraction of outliers increases or as the number of inliers decreases. Somewhat counter-intuitively, we obtain somewhat low accuracy with small fractions of outliers when SIFT features are used. This occurs because our test data in this segment is small (only 30 pairs).
>
> Q1. Meaning of “noise-free”:
>
> We apologize for this confusion. Existing methods use the essential matrix estimated *in inference* to correct the positions of key-points. Our network, in contrast, learns to denoise these positions independently from the estimated essential matrix, as we further explained in W2 above. Of course, the noise-free points used *in training* are obtained by applying geometry (triangulation). We will rephrase this statement
>
> Q2. Model size and Inference time:
>
> We address this question in Table R.1. While our model uses more parameters than NCMNet and BCLNet, which use graph attention architectures, it is 4-6 times faster than these methods and consumes less GPU memory at inference. We further compare our runtime to RANSAC-based methods. With 100K iterations, RANSAC is significantly slower than our method. We note that RANSAC is implemented in CPU. Kornia[1] introduced a GPU RANSAC implementation for fundamental matrix estimation. We tested their resource usage on the same data to demonstrate the potential of GPU implementations for RANSAC. Using a batch size of 10000 samples, their model runtime and maximum GPU memory usage were 40.94ms and 414.66MB, respectively, which are 4 times higher than our model.
>
> [1] Riba, Edgar, et al. "Kornia: an Open Source Differentiable Computer Vision Library for PyTorch.” Winter Conference on Applications of Computer Vision. 2020.

---

> > ### Comment · Reviewer_HiXm · 2024-08-08
> > **Thanks for taking time to write rebuttal.**
> >
> > Reply W1: Thanks for providing the graph. What happens for image pairs with high pose error? (or what are some highest key-point movement you see between before and after denoising?)  What I am trying to understand is what the noise-prediction block is learning? Is it learning through image gradient / feature gradients? Or is there something else going on here.
> >
> > Reply W2: Thanks, all clear.
> >
> > Reply W3: Make sense. This is a good insight, if possible I'll encourage authors to make space for it in the paper / appendix.
> >
> > Reply Q1: Thanks, this clarifies.
> >
> > Reply Q2: Amazing! I think this is very important. 11ms is highly desirable. Also, low memory usage shows that this method can also be used on smaller memory constrained devices.
> >
> > Overall, I am satisfied with authors response and the paper does show practical and novel algorithm for consensus learning. I will keep my rating to 8.

---

> > > ### Author Response · Authors · 2024-08-12
> > >
> > > Thank you for these comments. We will modify the paper accordingly.
> > >
> > > W1. With high pose error, we see nearly no correction at all. Perhaps we can answer your question regarding the noise prediction block with an informal discussion. Our noise prediction block is a function of a match $x_i$ and the full set of input matches $X$. This function is trained to minimize a loss that depends on $x_i$ and the ground truth essential matrix $E$, and we expect it to produce the maximal a postriori noise vector given the set $X$. Here the prior encodes the noise distribution of the feature extractor. For SIFT, this may be an isotropic Gaussian distribution. The set $X$ imposes a distribution over possible essential matrices, and those, in turn, narrow down the likely directions and magnitudes of the predicted error.

---

> > > > ### Comment · Reviewer_HiXm · 2024-08-12
> > > >
> > > > Thanks for clarifying. This would have been amazing result / ablation to add to the paper. However, I have no further questions.

---

### Official Review · Reviewer_THYF · 2024-07-14

**Soundness:** 3
**Presentation:** 4
**Contribution:** 3
**Rating:** 7
**Confidence:** 4

**Summary:**

The authors propose a method to tackle the traditional computer vision problem of estimating the essential matrix between two camera views of the same scene based on a set of point matches. The method distinguishes between inlier / outlier matches, and explicitly models the displacement noise in the inlier matches using a “Noise Aware Consensus Network”. The model is trained in two stages: first, the model sees only noise-less inlier matches with outlier matches, and secondly real world blends of noisy inlier and outlier matches. The authors show that this framework is able to compute accurate essential matrices with a variety of different image descriptors over different datasets.

**Strengths:**

1. The paper presented convincing performance as compared to the baselines, especially in the cross-scene / cross dataset settings. This attribute is perhaps most important for practical applications, and shows that the explicit noise reasoning scheme appears to work well.

2. The ablation studies shed light on the contributions of the 2-stage training and keypoint denoising steps. These will help guide future research.

3. The architecture and training details are quite clear. Together with the authors’ promise of releasing the code, this work is a meaningful contribution to the research community.

4. The paper is easy to read and navigate; different parts of the design are motivated clearly.

**Weaknesses:**

1. It would be important to understand how the capacity of the NACNet architecture compares with the baseline methods - while it indeed appears simpler and more streamlined than pipelines such as BCNet, it would be important to understand whether the complexity has simply been wrapped into higher capacity models.

2. It seems that correspondence pruning should have a large impact on the performance of the model as well, but not a lot of attention is given to this area (e.g. in ablation studies); it seems that only some qualitative results are shown in Fig 4 and in the appendix. However, what is the actual performance of the prediction of Y_i (inlier / outlier labeling) in the different settings (in-scene, cross-scene, cross-dataset, etc)? Does this have a large impact on the performance?

**Questions:**

It would be great to read the authors’ responses to the issues raised in the weaknesses section.

**Limitations:**

The authors discussed the limitations of their work well with respect to the requirements in the Checklist.

---

> ### Author Rebuttal · Authors · 2024-08-07
>
> We thank the reviewer for these comments.
>
> W1. Network capacity and complexity:
>
> We address this question in Table R.1. While our model uses more parameters than NCMNet and BCLNet, which use graph attention architectures, it is 4-6 times faster than these methods and consumes less GPU memory at inference.
>
> W2. Pruning and classification evaluation:
>
> We discuss the effect of correspondence pruning in our paper (ablation is mentioned in Section 4.4, lines 246-255). Similar to MGNet, we see that pruning has a slight negative effect (~-0.8%) on accuracy. We further attach Table R.3 to show classification accuracy. It can be seen that our model achieves higher F1 scores compared to previous methods, possibly explaining our overall improved regression accuracy.

---

> > ### Comment · Reviewer_THYF · 2024-08-08
> >
> > Thank you for the clarifications. It is great to see that the proposed model is more efficient than the baselines, and that the inlier/outlier classification seems to work well. I have no other concerns. Consequently, I've also adjusted the score up by 1 point.

---

> > > ### Author Response · Authors · 2024-08-12
> > >
> > > Thank you for this positive feedback!

---

### Author Rebuttal · Authors · 2024-08-07

We thank the reviewers for their comments.

We addressed each of your questions individually. Please note the attached pdf.

---

### Decision · Program_Chairs · 2024-09-25

**Decision:**

Accept (poster)

**Comment:**

All reviewers were initially positive and remained positive after the rebuttal. Reviewers commented on the good results, and as an application paper, this paper is worth noting. The AC thus follows this unanimous recommendation and recommends accepting the paper. A note that was made by one of the reviewers during the discussions was that the paper could have had a larger impact, if the paper evaluated beyond essential matrices, e.g., other robust estimation problems. But without it, the AC agrees that the paper's impact could be limited as while this is a plausible hypothesis, it remains untested.